# The WT1–BASP1 complex is required to maintain the differentiated state of taste receptor cells

Yankun Gao[1], Debarghya Dutta Banik[1], Mutia M Muna[1], Stefan GE Roberts[1,2], Kathryn F Medler[1]

**WT1 is a transcriptional activator that controls the boundary between multipotency and differentiation. The transcriptional cofactor BASP1 binds to WT1, forming a transcriptional repressor complex that drives differentiation in cultured cells; however, this proposed mechanism has not been demonstrated in vivo. We used the peripheral taste system as a model to determine how BASP1 regulates the function of WT1. During development, WT1 is highly expressed in the developing taste cells while BASP1 is absent. By the end of development, BASP1 and WT1 are co-expressed in taste cells, where they both occupy the promoter of WT1 target genes. Using a conditional BASP1 mouse, we demonstrate that BASP1 is critical to maintain the differentiated state of adult taste cells and that loss of BASP1 expression significantly alters the composition and function of these cells. This includes the de-repression of WT1-dependent target genes from the Wnt and Shh pathways that are normally only transcriptionally activated by WT1 in the undifferentiated taste cells. Our results uncover a central role for the WT1–BASP1 complex in maintaining cell differentiation in vivo.**

## Introduction

The WT1 transcription factor plays a critical role in the development and maintenance of multiple organs and tissues (Hastie, 2017). In particular, WT1 null mice display complete agenesis of the kidneys, gonads, adrenal glands, and spleen. WT1 is also needed for tissue maintenance in the adult, with these sites having some overlap with developmental targets as well as additional organs (Chau et al, 2011). WT1 can either drive cell proliferation or promote differentiation, but the mechanisms involved in this dichotomy are not clear (Toska & Roberts, 2014; Hastie, 2017).

WT1 often acts in concert with a transcriptional cofactor, BASP1. BASP1 binding switches the function of WT1 from an activator to a repressor (Toska & Roberts, 2014) and regulates the ability of WT1 to control differentiation in several model cell lines, including kidney podocyte cells (Green et al, 2009), epicardial cells (Essafi et al, 2011),

and blood cells (Goodfellow et al, 2011). Recent work has also shown that in the absence of BASP1, WT1 has an important role in maintaining multipotency. BASP1 blocks this function and is associated with driving iPSCs to differentiate (Blanchard et al, 2017). Thus, BASP1 is a critical regulator of WT1 function.

WT1 null mice also have developmental defects in several sensory tissues, including the retinal ganglion cells (Wagner et al, 2002), olfactory epithelia (Wagner et al, 2005), and, as shown by us, peripheral taste cells (Gao et al, 2014). A distinctive feature of peripheral taste cells is that they are continuously replaced throughout an organism's lifetime (Barlow & Klein, 2015), which generates a need for constant remodelling of these cells. We found that WT1 and BASP1 are expressed in adult taste cells, but their roles are currently unknown. Based on its function in other cell types, we hypothesized that the WT1/BASP1 complex contributes to the taste renewal process.

Taste receptor cells originate from Keratin 14 (Krt14+)–expressing progenitor cells that become either non-taste epithelium or postmitotic precursors that express sonic hedgehog (Shh+). These postmitotic Shh+ cells further differentiate into functional taste cells that express Keratin 8 (Krt8). Krt8 is highly expressed in the mature taste cells, which are located in taste buds present in the oral cavity. These cells are divided into one of three groups (type I, II, or III), which are based on their physiological function as well as the expression of specific markers and anatomical features (Liu et al, 2013; Barlow & Klein, 2015). The taste system is unique among most neuronal systems in that it undergoes constant cell renewal (Barlow, 2015). Differentiated taste receptor cells are housed in the taste bud for 8–12 d on average before being replaced by newly differentiated taste cells (Perea-Martinez et al, 2013). Thus, the taste bud is a dynamic grouping of a heterogeneous population of taste cells that have different functions within the bud. At any given time, the taste receptor cells within a particular bud are at different stages of their life span, including immature cells through to mature, fully differentiated cells.

The current understanding of this taste cell renewal process is far from complete. It is clear that both the Shh and Wnt/$\beta$-catenin signaling pathways regulate the specification of taste cell fate and are required for taste cell differentiation (Castillo et al, 2014; Gaillard et al, 2015; Gaillard et al, 2017). However, the underlying

---

[1]Department of Biological Sciences, University at Buffalo, Buffalo, NY, USA   [2]School of Cellular and Molecular Medicine, University of Bristol, Bristol, UK

Correspondence: kmedler@buffalo.edu; Stefan.Roberts@bristol.ac.uk

mechanisms regulating Wnt and Shh signaling in adult taste cells during this process are still unknown. The goal of this study was to analyze the role of BASP1 within taste cell renewal. We find that deletion of *BASP1* in differentiated cells leads to their reduced function, a loss of several cell type markers normally found in mature cells, and the up-regulation of WT1 target genes that are primarily expressed in the progenitor cells. Our findings reveal that the WT1–BASP1 complex plays a central role in the maintenance of the differentiated state in this system.

## Results and Discussion

Our previous work identified a key role for WT1 in the development of the peripheral taste system, specifically the circumvallate (CV) papillae (Gao et al, 2014). The CV papillae are an epithelial specialization located on the back of the tongue, which houses hundreds of taste buds. Because BASP1 is a critical transcriptional cofactor for WT1, we sought to determine if it acts with WT1 during the development of this papillae type. We analyzed BASP1 expression at different developmental stages during CV formation using double label immunohistochemistry with either Krt8+ (a marker of the developing taste cells; Fig 1A) or GAP43 (a marker that identifies the gustatory nerve, which will innervate the emerging taste cells; Fig 1B). Negative control data for the secondary antibodies are shown in Fig S1A.

At E12.5 and E15.5, BASP1 is present in the developing gustatory nerve (GAP43+) but is not co-expressed with Krt8 in the placode that is developing into the CV papillae. At P1, we began to detect some overlap in the BASP1 and Krt8 expression in the taste buds. At P4 and P7, BASP1 expression is present in both the Krt8+ cells of the taste buds and the GAP43+ gustatory nerve but its expression is more predominant in the taste buds and reduced in the gustatory nerve. The peripheral taste system is not fully functional at this time but continues to develop after birth (Barlow, 2015). By adulthood (P30 or later), BASP1 expression can no longer be detected in the GAP43+ gustatory nerve but is highly expressed in the differentiated taste buds. Thus, unlike WT1 (Gao et al, 2014), BASP1 is not expressed in the Krt8+ cells during CV development but is switched on around birth in these cells.

The adult CV papillae houses hundreds of taste buds that are innervated by gustatory nerves (Fig 2A). Each taste bud contains 50–150 taste receptor cells, which are identified as type I, II, or III based on their morphology and known functions. Taste receptor cells are derived from the surrounding progenitor cells that express Krt14+, which transition to a postmitotic stage (Shh+ cells) before differentiation into the specific taste receptor cell types (I, II, and III) that express Krt8 (Barlow & Klein, 2015).

Adult mouse taste cells express WT1 (Gao et al, 2014) and we find that it shows a large degree of cell overlap with BASP1 in the taste buds (Fig 2B). Negative control labelling is shown in Fig S1B. Both WT1 and BASP1 exhibit nuclear and cytoplasmic localization in the taste cells as assessed by comparison with DAPI (Fig 2C). To identify the specific cell types that express WT1 and BASP1, we performed immunohistochemistry using cell type–specific markers. NTPDase2 labels type I cells and has strong overlap with WT1 expression

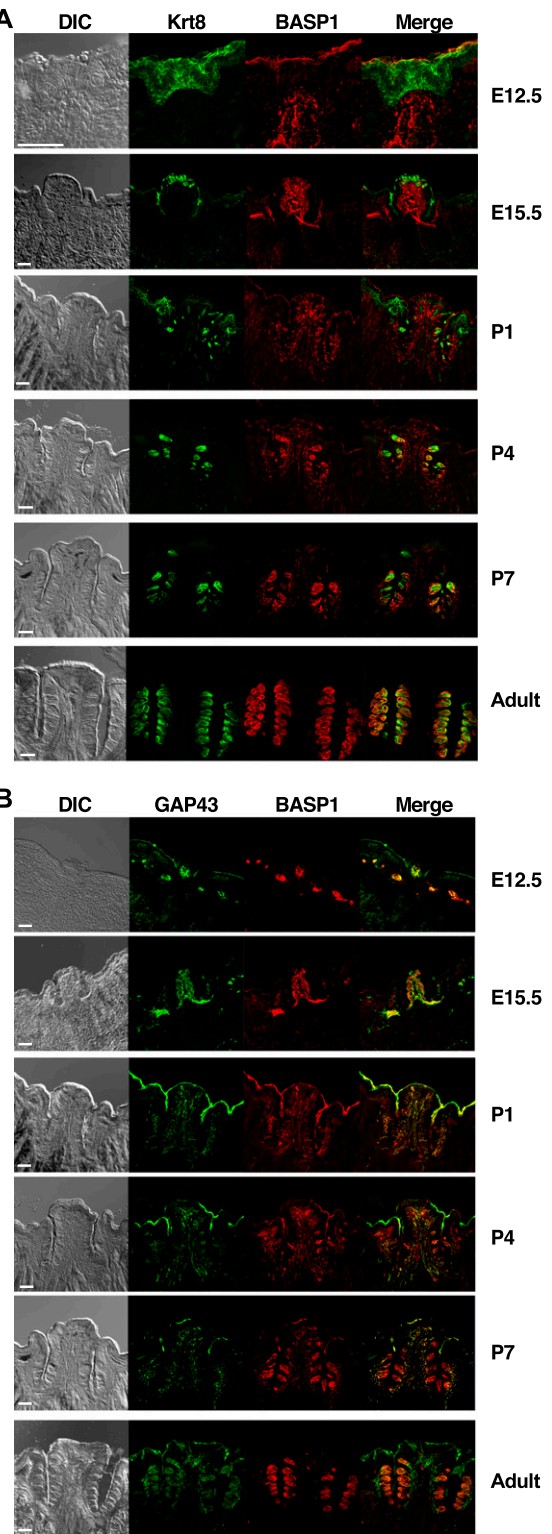

**Figure 1. BASP1 is expressed at the terminal stages of CV development.**
**(A)** Immunohistochemistry of CV at different stages of development (indicated at right). BASP1 is red and Krt8 is green, and merged image is shown. DIC is shown at the left. Adult mice are at least 30 d post birth. **(B)** As in part (A) except that BASP1 (red) and GAP43 (green) were detected. For all images, a stack of five slices (1 μm each) is shown. Scale bar is 50 μm.

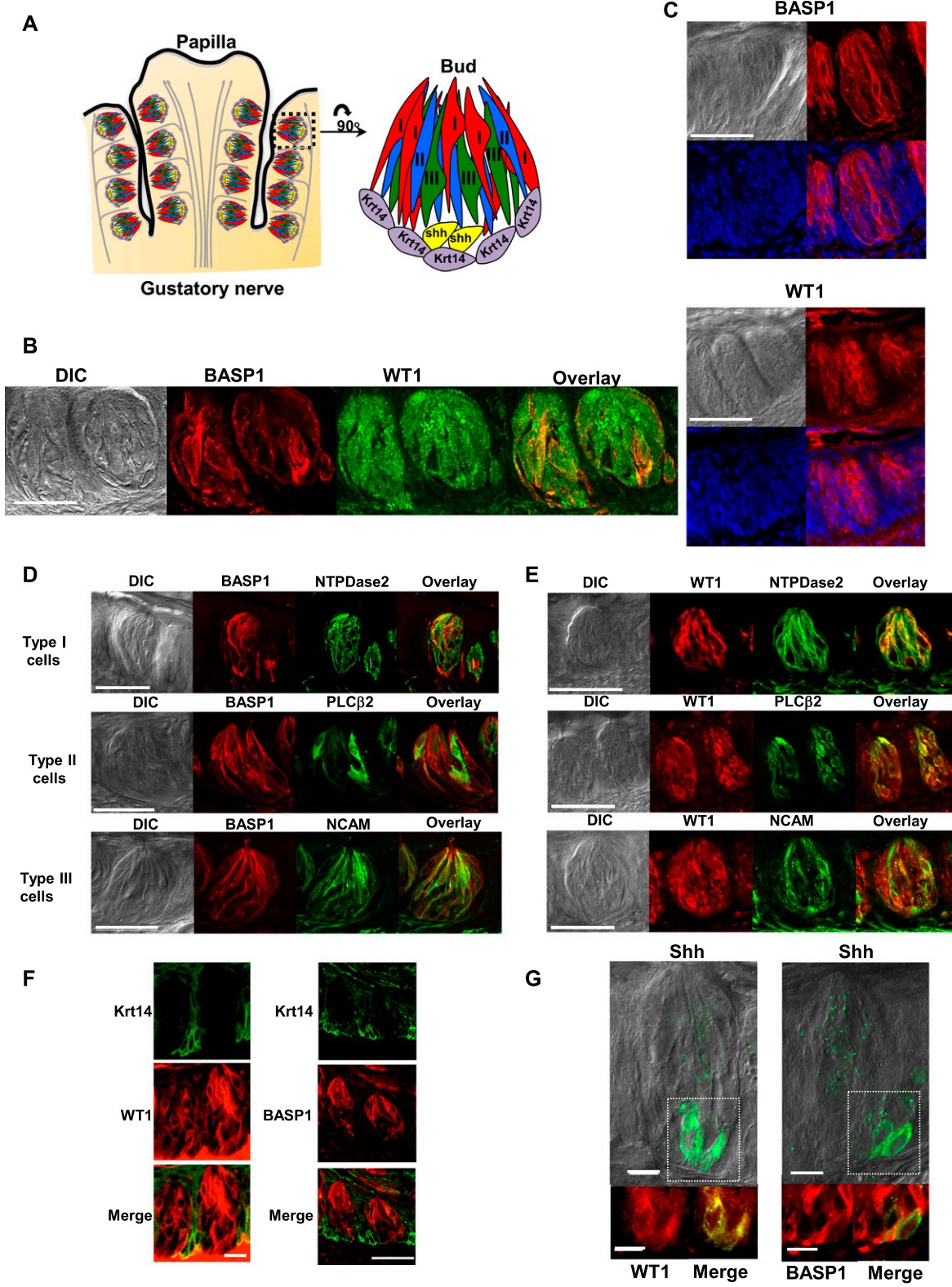

**Figure 2. WT1 and BASP1 are expressed in overlapping cell types of the adult CV.**

**(A)** Diagram of CV papillae showing the taste buds (comprising 50–150 cells) and innervation by the gustatory nerve. A single bud is shown at right containing the differentiated type I, II, and III cells, and the precursor K14+ and Shh+ cells. **(B)** WT1 and BASP1 are expressed in adult taste buds. Immunohistochemistry of CV papillae to detect BASP1 (red) and WT1 (green) with merged image. Scale bar is 50 $\mu$m. **(C)** Nuclear and cytoplasmic localization of WT1 and BASP1. BASP1 (top) and WT1 (bottom) are shown merged with DAPI staining. Scale bar is 50 $\mu$m. **(D)** Immunohistochemistry of CV taste buds to detect BASP1 (red) and either NTPDase2 (green; type I

(Fig 2E) but does not overlap with BASP1 (Fig 2D). In the type II (PLCβ2+) and III (NCAM+) cells, both WT1 and BASP1 are expressed (Fig 2D and E). Thus, WT1 is expressed in all three types of differentiated taste cells, whereas BASP1 expression is restricted to the type II and III cells. Type II and III cells are the transducers of taste stimuli, whereas type I cells are thought to function as support cells similar to glial cells in the central nervous system (Finger & Simon, 2000).

We also found that WT1, but not BASP1, is expressed in the progenitor (Krt14+) cells (Fig 2F). This finding suggests that WT1 is functioning very early in the cell renewal process during the proliferation and formation of new taste cells. This is consistent with WT1's function in the CV during embryonic development (Gao et al, 2014) as well as its role in iPSCs (Blanchard et al, 2017). WT1 is also found in the postmitotic progenitor (Shh+) cells that will ultimately differentiate into the mature taste cells (Fig 2G), whereas BASP1 is expressed in a subset of these Shh+ cells (n = 7 of 27 cells taken from six taste buds from four mice). This raises the possibility that BASP1 expression is initiated as the Shh+ cells undertake the final transition to terminal differentiation. Taken together, our results demonstrate that WT1 is expressed in both the progenitors and differentiated taste cells, whereas BASP1 expression is generally limited to the differentiated cells.

Our next experiments measured the effects of BASP1 on taste cell function. We crossed a floxed BASP1 mouse (Fig S2) with a Krt8-Cre-ER mouse to delete BASP1 in the fully differentiated Krt8-expressing taste cells. Mice were treated with tamoxifen for 8 d, euthanized a week later, and immunohistochemistry was performed to determine the level of BASP1 reduction in the taste buds (Fig 3A). We found that BASP1 was significantly reduced but not entirely eradicated after tamoxifen treatment. As mentioned above, the average half-life of differentiated taste cells is 8–12 d (Hamamichi et al, 2006; Perea-Martinez et al, 2013), and so we expect that some of the taste cells will be replaced during the experiment. Because Krt8 is only expressed in the fully differentiated taste cells (and not the precursor Krt14+ or Shh+ cells), we predict that there may be some limited recovery of BASP1-positive cells within the taste bud. RNA was also prepared from isolated taste cells to confirm that BASP1 mRNA in taste cells was significantly reduced (Fig 3B, left graph). The specific deletion of the BASP1 gene in Krt8-expressing cells (Krt8-BASP1-KO; KO) did not lead to any obvious change in the taste bud morphology compared with controls (CTLs) and the number of taste buds in the CV papillae of KO mice was not significantly different from CTL mice (Fig 3B, right graph). Furthermore, the expression levels of Krt8 in the taste buds were not altered in the KO mice (Fig 3C).

To evaluate if the loss of BASP1 affected physiological function, we isolated taste cells and compared the taste-evoked calcium responses between KO and CTL mice using live cell imaging. Two taste stimuli were used to activate type II cells: sweet (20 mM

sucralose) and bitter (5 mM denatonium benzoate, Den). Potassium chloride (50 mM, HiK) was used to depolarize type III taste cells and activate voltage-gated calcium channels. Using chi-square frequency analysis (Preacher, 2001), we measured a significant reduction in the overall number of responsive cells for all three stimuli in KO mice compared with CTLs (Fig 3D). Moreover, the remaining calcium signals for denatonium and sucralose were significantly reduced in amplitude (percent increase over baseline) in KO mice compared with CTLs (Fig 3E). Representative traces are shown in Fig S3. Taken together, these data reveal that the ability of the taste cells to respond to stimuli is significantly impaired when BASP1 expression is ablated.

Our results so far demonstrate that the expression of BASP1 in differentiated taste cells is required for their normal function and that the knockout of BASP1 significantly reduces their ability to respond to stimuli. We also measured the effect of BASP1 deletion on the expression of specific cell type markers. Deletion of BASP1 (Fig 4A) caused a significant increase in the expression of the type I cell marker, NTPDase2 (Fig 4B). We did not detect BASP1 in the type I cells of CTL mice (Fig 2D), suggesting either that a low level of BASP1 in type I cells normally acts to modulate NTPDase2 expression or that deletion of BASP1 in other Krt8+ cells affects the properties of type I cells. Conversely, loss of BASP1 in Krt8+ cells caused a reduction of both PLCβ2 (type II cell marker; Fig 4C) and NCAM (type III cell marker; Fig 4D) expression. The reduction in PLCβ2 and NCAM expression was not due to a decrease in the overall number of type II and III cells (Fig 4E), but was instead due to the lower expression of these proteins in the cells (Fig 4C and D, graphs). Not all type II and type III functional proteins were affected by BASP1 deletion as gustducin (type II marker) and pgp9.5 (type III marker) expression levels were maintained in the KO mice (Fig S4). The reduced expression of PLCβ2 and NCAM suggests a loss of cell specialization and is consistent with the impaired responses of the taste cells that we observed (Fig 3). Taken together, these data (Figs 3 and 4) suggest that loss of BASP1 in the differentiated taste cells leads to a significant alteration in their properties.

Our previous work identified the LEF1 and PTCH1 genes as direct targets for WT1 regulation during the development of the CV papillae (Gao et al, 2014). These genes encode components of the Wnt and Shh signaling pathways, respectively, that are important during the development of the peripheral taste system. These signaling pathways also play a central role during the early stages of the cell renewal process in the precursor cells (Krt14+ and Shh+ cells) but are generally switched off in the fully differentiated taste cells (Barlow & Klein, 2015). To determine if LEF1 and PTCH1 expression were affected by the loss of BASP1 in the differentiated taste cells, we prepared RNA from isolated taste cells of either KO or CTL mice. qPCR was used to measure the levels of LEF1 and PTCH1 mRNA relative to control GAPDH mRNA (Fig 5A) in the KO and CTL cells. Deletion of BASP1 in the Krt8-positive taste cells led

cells), PLCβ2 (green; type II cells) or NCAM (green; type III cells). DIC images are shown to the left of each with merged images of the labelling shown at right. Scale bar is 50 μm for each. **(E)** As in part (D) except that WT1 (red) was detected instead of BASP1. **(F)** Left panels: immunohistochemistry of CV taste buds to detect WT1 (red) and Krt14 (green). Merged image is shown at the bottom. Right panels: immunohistochemistry of CV taste buds to detect BASP1 (red) and Shh (green). Merged image is shown at bottom right. Scale bar is 10 μm for WT1 and 50 μm for BASP1. **(G)** As in part (D) except that Shh (green) was detected instead of Krt14. Scale bar is 10 μm. For all images, a stack of five slices (1 μm each) is shown.

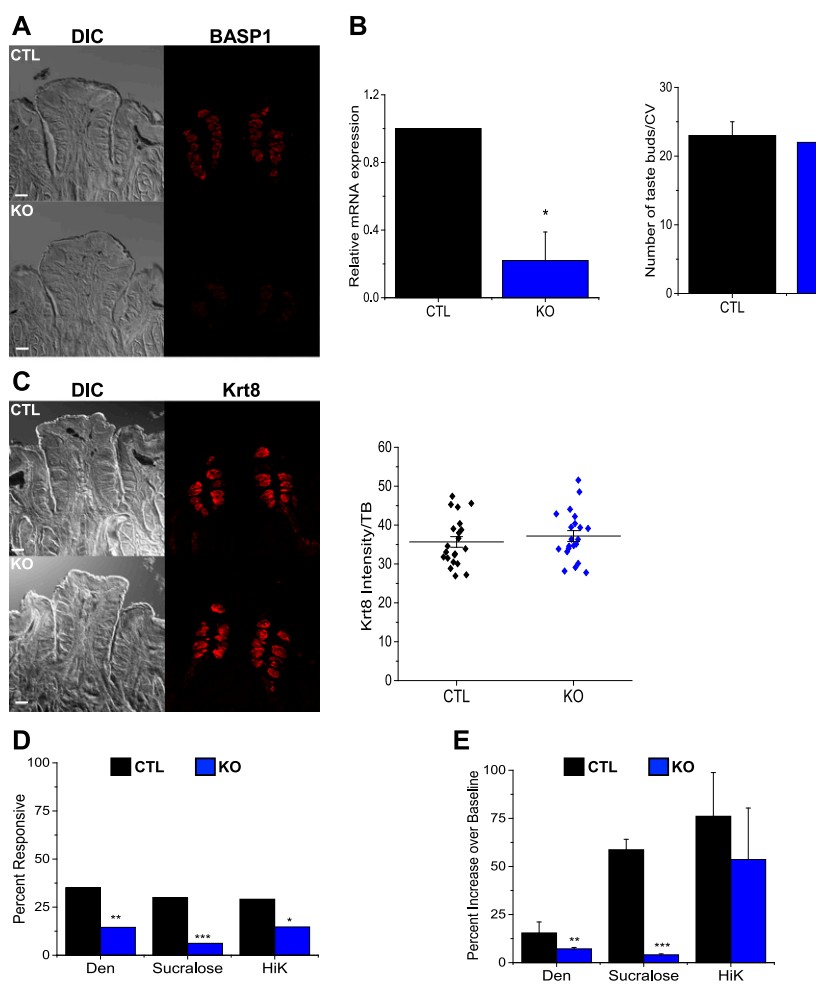

**Figure 3. Knockout of *BASP1* in the adult CV affects the stimulus responses of taste cells.**
**(A)** Immunohistochemistry of CV papillae to detect BASP1 (red) in control (CTL) and Krt8-BASP1-CRE (KO) mice treated with tamoxifen. Scale bar is 50 $\mu$m. **(B)** Left graph: qPCR was used to determine BASP1 expression relative to GAPDH in RNA isolated from CTL (black bar) and KO (blue bar, KO) mice (*$P < 0.05$). Right graph: taste buds were stained with DAPI and the number of cells per bud were counted in CTL and KO mice (n = 3 mice, seven buds per mouse for each). Mean with SD are reported. **(C)** As in part (A) except that staining was with anti-Krt8. At right is the quantitation of immunoreactivity per taste bud using ImageJ. Horizontal bars represent mean average intensity. For all images, a stack of five slices (1 $\mu$m each) is shown. **(D)** Chi-square analysis compared the overall number of responsive taste cells in CTL and KO mice. The percentage of responsive taste cells is shown (***$P < 0.001$; **$P < 0.01$; *$P < 0.05$) for 5 mM denatonium (CTL: n = 150 cells, five mice; KO: n = 145 cells, three mice), 20 mM sucralose (CTL: n = 150 cells, five mice; KO: n = 145 cells, three mice), and 50 mM potassium chloride (HiK) (CTL: n = 135 cells, five mice: KO: n = 129 cells, three mice). **(E)** The response amplitudes (percent increase over baseline) for the responsive cells from part (D) were analyzed and compared for the CTL and KO mice. Mean with SD are reported along with a *t* test (***$P < 0.001$ and **$P < 0.01$). Representative traces are shown in Fig S3.

to a significant increase in both *LEF1* and *PTCH1* expression compared with CTLs.

We then determined if WT1 and BASP1 directly localize to the promoter regions of the *LEF1* and *PTCH1* genes in these cells. Taste cells were isolated from the CV papillae of either CTLs or KO mice and subjected to chromatin immunoprecipitation (ChIP) using WT1, BASP1, or control (IgG) antibodies. The results are expressed as fold enrichment at the *LEF1* and *PTCH1* promoters over a control genomic region (Fig 5B). Whereas WT1 was localized at the *LEF1* and *PTCH1* promoters in both CTL and KO cells, BASP1 was only bound to the promoters of the *LEF1* and *PTCH1* genes in CTL cells and its binding was lost in the KO cells. Based on these data, we conclude that the WT1–BASP1 complex is directly repressing the transcription of *LEF1* and *PTCH1* in the differentiated taste cells. When BASP1 is knocked out, the mRNA levels of both *LEF1* and *PTCH1* are significantly up-regulated.

Immunohistochemical analysis evaluated the effects of *BASP1* knockout on the expression levels of LEF1 and PTCH1 proteins in the CV papillae compared with CTLs (Fig 5C). LEF1 and PTCH1 are normally only expressed at low levels in the differentiated cells with higher expression levels in the progenitors (Miura et al, 2001;

Gaillard et al, 2017). Our data demonstrate that knockout of *BASP1* in the differentiated taste cells (Krt8+) caused an up-regulation in the expression of both LEF1 and PTCH1 throughout the CV papillae when compared with CTLs (Fig 5C). Taken together, the results in Fig 5 demonstrate that WT1–BASP1 acts as a transcriptional repressor complex in taste cells that normally inhibits the expression of Wnt and Shh signaling components in the differentiated cells.

Our previous work reported that WT1 regulates the expression of *LEF1* and *PTCH1* genes during the embryonic development of the peripheral taste system, from very early stages through birth (Gao et al, 2014). We have now shown that BASP1 is not expressed during the embryonic development of the peripheral taste system but is turned on shortly after birth during its final developmental steps. The mechanism by which BASP1 is switched on at this time remains to be determined, but our data suggest that it coincides with the conversion of WT1 from a transcriptional activator to a repressor. WT1, but not BASP1, is expressed in the progenitor K14+ cells, which constitutively express LEF1 and PTCH1. In the K14+ cells, these signaling pathways direct their differentiation to the postmitotic, undifferentiated, Shh+ cells, which further differentiate into the functional taste cells (Kapsimali & Barlow, 2013; Barlow & Klein,

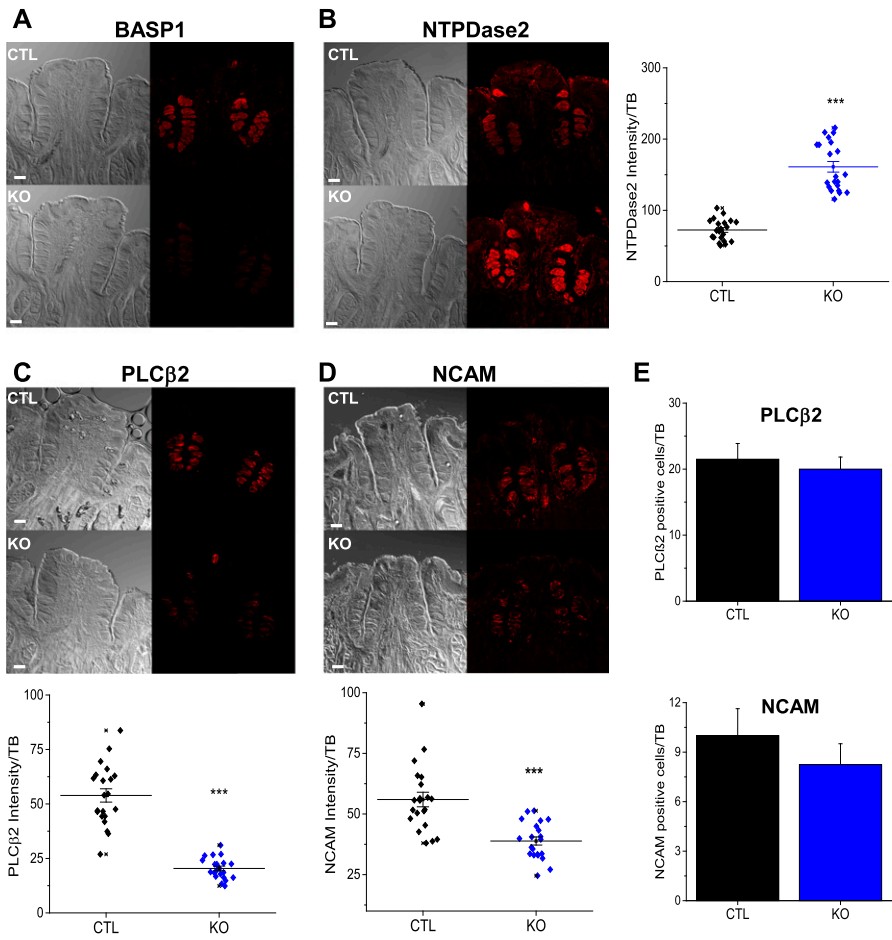

**Figure 4. Knockout of *BASP1* in the adult CV leads to a disruption of type II and type III cells markers.**
**(A)** Immunohistochemistry of CV taste buds using anti-BASP1 (red) in CTL and KO mice that had been treated with tamoxifen. Scale bar is 50 $\mu$m. **(B)** As in part (A) except that NTPDase2 (type I marker) was tested. A plot of signal intensity per taste bud is shown at right. Horizontal bars represent mean intensity (***$P < 0.001$ $t$ test). **(C)** As in part (A) except that anti-PLC$\beta$2 (type II marker) was used. A plot of signal intensity per taste bud is shown below. Horizontal bars represent mean intensity (***$P < 0.001$ $t$ test). **(D)** As in part (A) except that NCAM labelling (type III marker) was measured. A plot of signal intensity per taste bud is shown below. Horizontal bars represent mean intensity (***$P < 0.001$ $t$ test). For all images, a stack of five slices (1 $\mu$m each) is shown. **(E)** The number of PLC$\beta$2– (above) and NCAM– (below) positive cells per taste bud in CTL and KO cells is shown. Seven buds from three mice for CTL and KO were analyzed for each antibody. Error bars are SD of the mean.

2015; Gaillard et al, 2015). The selective expression of BASP1 in a subset of Shh+ cells suggests that BASP1 binding may be needed to drive the Shh+ cells to differentiate into either type II or type III cells, whereas WT1 alone is sufficient to cause differentiation of type I cells.

In cell culture systems, the absence of BASP1 allows WT1 to recruit the transcriptional coactivator CREB-binding protein (Essafi et al, 2011; Toska et al, 2014). BASP1 displaces CREB-binding protein from WT1 and promotes the recruitment of HDAC1 (Toska et al, 2012; Toska et al, 2014). These events lead to histone H3K9 deacetylation and H3K27 trimethylation (Essafi et al, 2011; Toska et al, 2012). Future studies will determine if similar chromatin regulatory mechanisms are used by BASP1 to repress transcription of the LEF1 and PTCH1 genes in taste cells.

Our results show that the sustained presence of the WT1–BASP1 complex is required to maintain taste receptor cells in their differentiated state. Such a mechanism is consistent with the recently reported roles of WT1 and BASP1 in stem cells in which WT1 promotes multipotency, whereas the WT1–BASP1 complex maintains the differentiated state and inhibits the induction of pluripotency (Blanchard et al, 2017). Several in vitro studies have demonstrated that BASP1 converts WT1 from a transcriptional activator to a repressor. This activator–repressor switch allows the WT1/BASP1 complex to regulate the differentiated state of the cell in vitro

(Toska & Roberts, 2014). Our data are the first demonstration that this dual-control mechanism of the WT1/BASP1 complex is critical for the differentiated state of a system in vivo.

# Materials and Methods

### Animals

Floxed BASP1 mice were generated in a C57BL/6 background by Cyagen Biosciences (Fig S2). Animals were cared for in compliance with the University at Buffalo Animal Care and Use Committee. BASP1fl$^{+/-}$ mice were mated with Krt8-Cre/ERT2$^+$ mice ([Zhang et al, 2012]; 017947; The Jackson Laboratory) to obtain BASP1fl$^{+/-}$Krt8-Cre/ERT2$^+$ mice. These offspring were bred to obtain BASP1fl$^{+/+}$Krt8-Cre/ERT2$^+$ mice, which were used as BASP1-KO group. The BASP1fl$^{-/-}$; Krt8-Cre/ERT2$^+$ or BASP1fl$^{+/+}$;Krt8-Cre/ERT2$^-$ littermates were used as wild-type CTLs. PCR primers used to genotype the conditional BASP1 knockout allele (see Fig S2): 5′-TGCCCTGCCTGCAGGTCAAT-3′ (F2); 5′-CCGAGTCTGTACAAAAGCCACC-3′ (R2). Tamoxifen was dissolved in corn oil mixed with ethanol (9:1 by volume) to a stock concentration of 20 mg/ml as previously described (Miura et al, 2014). BASP1-KO and CTL mice were gavaged with tamoxifen (100

**Figure 5. The WT1–BASP1 complex represses *LEF1* and *PTCH1* expression in the differentiated cells of the CV.** **(A)** Quantitative PCR (qPCR) was used to detect *LEF1* (above) and *PTCH1* (below) expression relative to *GAPDH* in mRNA isolated from taste cells of CTL and KO mice. Means (of three independent experiments) with SD are reported (*$P < 0.05$, $t$ test). **(B)** ChIP analysis of isolated taste cells from adult mice. WT1, BASP1, or control (IgG) antibodies were used. After normalization to input DNA, fold enrichment at the *LEF1* (top) or *PTCH1* (bottom) promoter regions over a control genomic region is shown in CTL and KO mice. Mean (of three independent experiments) with SD are reported (*$P < 0.05$, ***$P < 0.001$; $t$ test). **(C)** Immunohistochemical analysis of LEF1 expression in the CV of CTL or KO mice (left panels). Expression levels of PTCH1 in the CV of CTL or KO mice (right panels). Scale bar is 20 $\mu$m.

mg/Kg body weight) every 24 h for eight consecutive days. Mice were euthanized and cells/tissues were collected 7 d after the last gavage. Primers used to genotype the Cre induced BASP1-KO allele (see Fig S2): 5′-CGAGAGATTTGTGATGTATGATGAGCAG-3′ (F1); 5′-CCGAGTCTGTACAAAAGCCACC-3′ (R2).

**Taste receptor cell isolation**

Taste receptor cells were harvested from CV papillae of adult mice as previously described (Hacker & Medler, 2008; Laskowski & Medler, 2009; Rebello & Medler, 2010; Szebenyi et al, 2010; Rebello et al, 2013; Dutta Banik et al, 2018). Briefly, mice were euthanized using carbon dioxide and cervical dislocation. Tongues were removed and an enzyme solution containing 0.7 mg/ml Collagenase B (Roche), 3 mg/ml Dispase II (Roche), and 1 mg/ml trypsin inhibitor (Sigma-Aldrich) was injected beneath the lingual epithelium. After the tongues were incubated in oxygenated Tyrode's solution for ~17 min, the epithelium was peeled from the underlying muscle and pinned serosal side up before it was incubated in Ca²⁺-free Tyrode's for ~25 min. The cells were removed from taste papillae using capillary pipettes with gentle suction and placed onto coverslips coated with Cell-Tak (Corning) for live cell imaging. If the isolated taste cells were to be used for either ChIP or mRNA analysis, they were aspirated into a centrifuge tube and spun at 13,000*g* for 5 min to pellet the cells before use.

**Live cell imaging**

All measurements of intracellular calcium ($Ca^{2+}$) were performed in isolated taste receptor cells from the CV papillae. The cells were loaded for 20 min at room temperature with 2 $\mu$M Fura2-AM (Molecular Probes; Invitrogen) containing Pluronic F-127 (Molecular Probes). Loaded cells were then washed under constant perfusion for 20 min. Taste stimuli (sweet: 20 mM sucralose and bitter: 5 mM denatonium) and high potassium (50 mM KCl) solutions were individually applied. The cells were visualized using an Olympus IX73 microscope with a 40× oil immersion lens and images were captured with a Hamamatsu ORCA-03G camera (Hamamatsu Photonics K.K., SZK Japan). Excitation wavelengths of 340 and 380 nm were used with an emission wavelength of 510 nm. Cells were kept under constant perfusion using a gravity flow perfusion system (Automate Scientific). Images were collected every 4 s using Imaging

Workbench 6.0 (Indec Biosystems). Experiments were graphed and analyzed using Origin 9.2 software (OriginLab). An evoked response was defined as measurable if the increase in fluorescence was at least two standard deviations above baseline.

Analysis of the taste responsiveness was performed using an interactive chi-square analysis with Yate's correction for continuity (Preacher, 2001). Significant differences were reported if $P < 0.05$. These data are a measure of the response frequencies (total number of responsive cells/total number of cells stimulated) and are reported as a percentage. We also measured the response amplitude to approximate the size of the peak response [((peak response value − baseline)/baseline)*100]. Changes in the response amplitude were identified by using a $t$ test to ascertain any significant differences between BASP1-KO and CTLs for the taste-evoked signals. Significant differences were reported if $P < 0.05$.

### Immunohistochemistry

Fluorescent immunohistochemistry was performed as described by Gao et al, (2014) except for experiments with LEF1 antibodies in which tongue sections were fixed for 10 s in 4% paraformaldehyde/0.1 M phosphate buffer (PB, pH 7.2) at room temperature. Since the labelling is difficult to distinguish for individual cells for NTPDase2 expression, fluorescence intensity was quantitated using ImageJ as previously described (Dutta Banik et al, 2018). Three mice were used for each group (seven taste buds for each mouse) were analyzed.

PTCH1 immunohistochemistry was as follows: mice were anesthetized with sodium pentobarbital (40 mg/kg; Patterson Veterinary) and then transcardially perfused with a 0.025% heparin solution in 1% sodium nitrite, followed by a 4% paraformaldehyde solution in 0.1 M phosphate buffer (pH 7.2). After perfusion, the tongues were removed and placed into 4% paraformaldehyde for 2 h, followed by 4°C overnight cryoprotection in 20% sucrose. On the following day, the tongues were frozen in O.C.T. Compound (Sakura Finertek USA), and 12-$\mu$m sections of the CV papillae were cut. The slides were incubated in 3% hydrogen peroxide blocking reagent for 5 min. After washing, the sections were incubated in serum-blocking reagent, avidin-blocking reagent, and biotin-blocking reagent for 15 min each. The sections were then incubated in PTCH1 or control antibodies for 2 h at RT before overnight incubation at 4°C. On the next day, the sections were washed 3 × 15-min each in buffer and incubated with biotinylated secondary antibody for 60 min, followed by 3 × 15-min washes in buffer. The sections were then incubated in HSS-HRP for 30 min, followed by 3 × 2-min washes in buffer. Binding of primary antibody to the sections was visualized using a diaminobenzidine tetrahydrochloride chromagen (R&D Biosystems).

Anti-Shh (sc-365112) and anti-gustducin (sc-395) were from Santa Cruz, anti-Krt8 from Developmental Studies Hybridoma Bank, anti-Lef1 from Signal Transduction (C12A5), anti-GAP43 from Aves (GAP43), anti-NTPDase2 from Ectonucleotidases-ab.com (CD39L1), anti-PLC$\beta$2 from GeneTex (GTX133765), anti-NCAM (AB5032), anti-PGP9.5 (AB108986) and anti-Ptch1 (AB53715) were from Abcam, and anti-Krt14 from BioLegend (905301). WT1 and BASP1 antibodies were raised against a peptide (WT1) or recombinant protein (BASP1) by Pacific Immunology and then purified by affinity chromatography. Antibody specificity was verified as described in Green et al (2009),

Gao et al (2014), and Toska et al (2014). All experiments were performed on a minimum of three mice for each.

All images were obtained using a Zeiss LSM 710 Confocal Microscope (Zeiss). Stacks consisting of 5–7 slices (1 $\mu$m each) were collected using Zen software (Zeiss) and images (compressed slices for each) were processed using Adobe Photoshop CS5 software adjusting only brightness and contrast. Any comparisons between CTL and KO mouse samples were performed side by side using the same experimental parameters.

### RNA and ChIP analysis

Total RNA was extracted from taste cells using the Nucleospin RNA XS kit (Clontech). Unamplified total RNA was DNase-treated and then reverse-transcribed using the Bio-Rad cDNA synthesis kit. Primers for analysis of *GAPDH*, *LEF1*, and *PTCH1* are described in Gao et al (2014). Samples were run in triplicate and at least three biological repeats were performed for each experiment. ChIP assays were performed as described before (Wang et al, 2010) using primers for *LEF1*, *PTCH1*, and control regions described by Gao et al (2014). Datasets are an average of three biological repeats with SD and significance calculated by $t$ test.

## Supplementary Information

## Acknowledgements

This work was funded by the NIH National Institute of General Medical Sciences (1R01GM098609 to KF Medler and SGE Roberts) and by Medical Research Council (MR/K001027/1 to SGE Roberts). We thank Alan Siegel and the UB North Campus Imaging Facility funded by NSF-MRI Grant DBI 0923133 for the confocal images.

### Author Contributions

Y Gao: formal analysis and investigation.
D Dutta Banik: formal analysis and investigation.
MM Muna: investigation.
SGE Roberts: conceptualization, formal analysis, supervision, funding acquisition, investigation, methodology, project administration, and writing—original draft, review, and editing.
KF Medler: conceptualization, formal analysis, supervision, funding acquisition, investigation, methodology, project administration, and writing—original draft, review, and editing.

### Conflict of Interest Statement

The authors declare that they have no conflict of interest.

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
