## [Reviewer comments · Life Science Alliance]

Life Science Alliance

The WT1-BASP1 complex is required to maintain the differentiated state of taste receptor cells

Yankun Gao, Debarghya Dutta Banik, Mutia Muna, Stefan Roberts, and Kathryn Medler

DOI: <https://doi.org/10.26508/lsa.201800287>

Corresponding author(s): Kathryn Medler, University at Buffalo

Review Timeline:	Submission Date:	2018-12-19
	Editorial Decision:	2019-02-15
	Revision Received:	2019-04-30
	Editorial Decision:	2019-05-17
	Revision Received:	2019-05-27
	Accepted:	2019-05-27

Scientific Editor: Andrea Leibfried

Transaction Report:

February 15, 2019

Re: Life Science Alliance manuscript #LSA-2018-00287-T

Dr. Kathryn F Medler
University at Buffalo
Biological Sciences
University at Buffalo
Cooke Hall
Buffalo, NY 14260

Dear Dr. Medler,

Thank you for submitting your manuscript entitled "The WT1-BASP1 complex is required to maintain the differentiated state of taste receptor cells" to Life Science Alliance. The manuscript was assessed by expert reviewers, whose comments are appended to this letter.

As you will see, the reviewers think that your conclusions are not sufficiently supported by the data provided. A rigorous analysis of the conditional knock-out is missing, the imaging data are not convincing, and a mouse-specific analysis of the anti-WT1 antibody is lacking. While addressing these concerns is feasible, the outcome is rather unclear at this stage. We nevertheless decided to invite you to submit a revised version of this work, should you be able to address the reviewers' concerns. So please consider your options carefully; acceptance will depend on receiving strong support from the reviewers on the revised version. Please let us know in case you'd rather withdraw your manuscript from consideration here.

The typical timeframe for revisions is three months. Please note that papers are generally considered through only one revision cycle.

Thank you for this interesting contribution to Life Science Alliance. We are looking forward to receiving your revised manuscript.

Sincerely,

B. MANUSCRIPT ORGANIZATION AND FORMATTING:

Reviewer #1 (Comments to the Authors (Required)):

Basp1 has been identified as a co-suppressor for the Wt1 tumor suppressor gene. Previous data

has confirmed the importance of Basp1 for Wt1 to act as a transcriptional suppressor, but the overall *in vivo* role(s) of Basp1 had not been addressed yet. In this manuscript Gao describe the analysis of a conditional knockout mouse model for Basp1 in the developing taste buds using Krt8-CreERT2. They describe the expression of Basp1 in the development of the taste buds in relation to Wt1 (and other markers) and find co-expression of the two towards to end of the developmental process. In the conditional knockout they find that the differentiated state is lost, and describe de-repression of two genes, Lef1 and Ptch1, that were previously described as targets for Wt1-mediated suppression in these cells and linking this to reactivation of two pathways (Wnt and Shh) that are essential in early stage of development.

The presented data is potentially extremely interesting from a general point of view (how is a differentiated state maintained) and from the Wt1 angle (what is the role of Wt1 in this process and how does this relate to the shift of the protein between activating and suppressing modes). However, there are some issues that need to be addressed before I could support publication of this work.

1. This is the first publication of the conditional Basp1 allele but the basic description of the model is almost completely absent. No data is shown confirming correct targeting of the endogenous locus, for this preferably Southern blots should be shown, or at least arm-spanning PCRs. All PCRs that are shown in fig EV1 B and C would give the same results in random integration clones / mice and are not sufficient.

2. This is especially relevant in the light of the fact that data in fig. 3 and 4 do not show complete loss of Basp1 at protein level. There is clearly remaining signal using antibody staining, albeit lower than in controls. This cannot be explained by incomplete activity of the CreERT2 allele per se or due to the tamoxifen dose used. In both cases one would expect a black and white difference on the single cell level, Basp1 should be expressed normal or be completely absent in individual cells. Instead we see reduced expression in all cells. There needs to be more convincing data on what the effect on Cre induction is on Basp1 expression.

3. There are also other issues with the antibody staining of Basp1 and Wt1. In fig 2A both are shown as clearly nuclear, although there is some additional signal for Wt1 in different parts of the sample. Is this specific signal? Is this part of the same cells (I'm not familiar with looking at taste buds unfortunately). However, in fig 2D both are clearly cytoplasmic instead of nuclear. It is known that Wt1 can be both, but this should not differ in the same cell type in different sections. I don't know if Basp1 has also been described to be cytoplasmic or nuclear. As it is now, these data are not convincing.

4. On page 5 the authors claim that Basp1 expression is switched on 'around birth'. However, this is based on samples from E15.5 (off) and P7 (on). This is an almost 2 week window. There is no basis for the claim this is 'around birth'. Moreover, it is more relevant how this happens in the developmental process than when this happens in time.

5. Fig. 3D lacks error bars.

6. At the moment it is not clear if the loss of Basp1 only results in de-repression of Lef1 and Ptch expression or if there is an active shift towards activation of the genes. This could be tested rather easily and provide much information on the mechanism behind this shift. Essafi et al showed that the difference in Wt1 acting as suppressor or activator is mirrored in the binding of Basp1 or CBP/p300 to the Wt1-bound enhancer element. Determining whether CBP/p300 binding increases

upon Basp1 loss would determine if there is an active shift towards an earlier stage or not.

7. I assume Krt8 is not specific for only the taste buds. Could the authors comment on (potential) phenotypes in other organs / cell types?

8. Has the expression of Basp1 been studied in other tissues, and if so, is its expression always linked to Wt1 or could Basp1 have additional Wt1-independent roles?

Reviewer #2 (Comments to the Authors (Required)):

In this manuscript titled "The WT1-BASP1 complex is required to maintain the differentiated state of taste receptor cells", the author reported that the WT1-BASP1 complex plays a critical role in maintaining taste cell differentiation in vivo using a conditional BASP1 mouse. This represents an advance in understanding taste cell differentiation in the gustatory field. This work is well done, and the data largely support their conclusions. Below are a few minor comments that should be addressed prior to acceptance of the work for publication.

WT1 appears to be expressed in nucleus in Fig. 1A, but not so obvious in Fig. 1B or 1D. I am wondering how specific the antibody against WT1 is. In the manuscript, the authors cited Toska et al for validation of antibody specificity. A bit more details should be provided to make this clear.

For Fig. 3D, is there a particular reason using Chi-square for statistical analysis? How many cells were sampled?

It is unclear from the method section about how RNA was extracted from taste cells. More details will help readers to know what proportions of the preparation are from taste tissue and from the surrounding non-taste tissue. How did the authors control this for their experiments?

The authors draw their conclusion in a large part by immunostaining. One caveat is whether the staining for the same marker was done side by side for sections from control and knockout mice. This potentially could influence the relative intensity of staining.

Reviewer #3 (Comments to the Authors (Required)):

This is an interesting study investigating the function of the WT1/BASP1 transcriptional regulators in taste bud cell renewal. The authors present data from mice that suggest WT1/BASP1 maintain differentiated taste cell fate by repressing Wnt and Hedgehog signaling factors cell autonomously, while WT1 expression in progenitors promotes WNT and Hedgehog signaling, allowing these cells to proliferate and generate new taste cells. The most interesting part of the study is that when BASP1 is knocked down in mouse taste bud cells, some markers of differentiated taste receptor cells are reduced as is the response of isolated taste cells to sweet and bitter taste stimuli. However much of the data presented are not convincing, especially the immunostaining in figures 1-3, as detailed below. Better descriptions of the methods, especially microscopy used in the paper,

is essential.

Figure 1 From the images included in A and B it is not possible to identify double labeling. The authors do not indicate if these micrographs were obtained using confocal or light microscopy. If the latter, then double labeling structures simply cannot be reliably determined. If the former, the authors need to show single optical sections or indicate how many optical sections are compressed in each image. Further in the methods, the authors do not include criteria they used to classify double immunostaining. The quality of the microscopy in panel B is poor, especially the top 2 panels of the right column are not of sufficient resolution.

Figure 2. comments for Figure 1 apply, re: clear information of microscopy and criteria for double immunostained cells is lacking. The authors state for Fig 2A that: " it (WT1) shows a large degree of colocalization with BASP1." Please provide quantification to support this statement. Same critiques for panels B and C. Panel E. How is nuclear localization of BASP1 determined? The authors need a nuclear counterstain to make this judgement. The authors state that 7/27 Shh+ cells had nuclear BASP1. How many taste buds from how many mice were used?

Figure 3. In the BASP1 conditional mice it appears that BASP1 expression is knocked down, not not knocked out, at both the protein and mRNA levels. The authors should use "knockdown" throughout. Please clarify if qPCR results reflect levels of BASP1 in the entire CVP epithelium or from plucked taste buds. Please indicate in the legend the number of mice and number of cells that contributed to the histograms in D and E

Figs 4 and EV3. BASP1 knockdown in Krt8 taste cells causes reductions in some markers of type II and III cells but not others, and that their expression data correlate with reduced responses of isolated taste cells to sweet and bitter and KCl depolarization. However, gustducin is expressed by bitter detecting cells in the CVP but expression levels are not changed. The data don't support the general conclusion the authors make at the end of the first paragraph on page 8.

Methods: the authors state that "antibody specificity was verified as described in Toska et al., 2014". This paper uses human cell lines, not mice and therefore they need to include sections processed with no primary as negative controls for immunostaining in mice.

Reviewer 1 comments

1. This is the first publication of the conditional *Basp1* allele but the basic description of the model is almost completely absent. No data is shown confirming correct targeting of the endogenous locus, for this preferably Southern blots should be shown, or at least arm-spanning PCRs. All PCRs that are shown in fig EV1 B and C would give the same results in random integration clones / mice and are not sufficient.

We apologize for the insufficient description of the data in Fig S1 (now Fig S2). We have now rectified this to clearly demonstrate that the endogenous *BASP1* gene has been replaced by the floxed *BASP1* construct. The primers used in Fig S2B and the left panel of Fig S2C are against the coding region and 3' homology arm of the *BASP1* allele and thus amplify both the wild type endogenous *BASP1* gene (producing a 345bp product) and the floxed *BASP1* transgene (producing a 467bp product). The gel in the left side of Fig S2C show a direct comparison of *BASP1*^{wt/wt} and *BASP1*^{fl/fl} confirming the replacement of the endogenous gene by the floxed *BASP1*. The gel on the right side of Fig S2C shows the same mice after treatment with tamoxifen. The primers (F1/R2) span the homology arms of *BASP1* and produce a 271bp product only when the floxed region containing Exon 2 is deleted. We have now shown the location of the priming sites in Figure S1A.

2. This is especially relevant in the light of the fact that data in fig. 3 and 4 do not show complete loss of *Basp1* at protein level. There is clearly remaining signal using antibody staining, albeit lower than in controls. This cannot be explained by incomplete activity of the CreERT2 allele per se or due to the tamoxifen dose used. In both cases one would expect a black and white difference on the single cell level, *Basp1* should be expressed normal or be completely absent in individual cells. Instead we see reduced expression in all cells. There needs to be more convincing data on what the effect on Cre induction is on *Basp1* expression.

The overlay of fluorescence and DIC images exaggerated the *BASP1* staining that is present after CRE induction. We have therefore presented the DIC image as a separate panel so that the structure of the tissue can be compared to the fluorescent image. We agree that there is still some residual fluorescence detected with *BASP1* antibodies following CRE induction but it is considerably reduced compared to the wild type mice. During our optimization experiments we found that 8 consecutive days of tamoxifen treatment followed by sacrifice after a further 7 days produced the optimal level of *BASP1* knockout. However, the average half-life of taste cells is 8-12 days and so some of the taste cells will be replaced during this time. Since *Krt8* is only expressed in the differentiated taste cells (and not the precursor *Krt14+* or *Shh+* cells) it is expected that there will be some limited recovery of *BASP1* positive cells within the taste bud. We have now explained this fully within the text. Based on the comments of Reviewers 1 and 2 we agree that a description of "BASP1 knockdown" is more appropriate to describe the data.

3. There are also other issues with the antibody staining of *Basp1* and *Wt1*. In fig 2A both are shown as clearly nuclear, although there is some additional signal for *Wt1* in different parts of the sample. Is this specific signal? Is this part of the same cells (I'm not familiar with looking at taste buds unfortunately). However, in fig 2D both are clearly cytoplasmic instead of nuclear. It is known that *Wt1* can be both, but this should not differ in the same cell type in different sections. I don't know if *Basp1* has also been described to be cytoplasmic or nuclear. As it is now, these data are not convincing.

In Fig 2A (now Fig 2C) the images shown were of entire taste papillae and the resolution was not sufficient to demonstrate subcellular localization. This figure was intended to demonstrate that both WT1 and BASP1 are localized to the taste buds within the papillae in adult mice. With hindsight we now appreciate that the manuscript did not provide sufficient information on the structure of the taste system and that the taste buds (which contain 50-150 cells each) look remarkably similar to individual nuclei in the low magnification immunohistochemistry. We have now provided a diagram in the new Fig 2A which describes the structure of the circumvallate papillae that house the individual taste buds. We hope this provides a better framework for the data shown in the rest of Fig 2. We have also included a higher magnification of an individual taste bud counterstained with DAPI for both BASP1 and WT1 in Fig 2C as well as a higher magnification of the BASP1/WT1 co-labeling (Fig 2B). Consistent with previous studies, WT1 and BASP1 show both nuclear and cytoplasmic staining.

4. On page 5 the authors claim that *Basp1* expression is switched on 'around birth'. However, this is based on samples from E15.5 (off) and P7 (on). This is an almost 2 week window. There is no basis for the claim this is 'around birth'. Moreover, it is more relevant how this happens in the developmental process than when this happens in time.

We agree with the Reviewer that the gap in the original timeframe shown was too large to make this statement. We have now included additional timepoints to illustrate that BASP1 expression within the taste cells occurs between P0 and P7. We agree with the Reviewer that the mechanism of BASP1 “switch on” is important. We now comment on this in the discussion section, including that this question forms a central part of our future studies.

5. Fig. 3D lacks error bars.

Figure 3D is a measure of the frequency of responses in the taste cells from the CTL and BASP1-KO mice. These data are presented as percentages and were analyzed using Chi-square analysis. Therefore, there should not be any error bars. Based on the reviewers' comments, we realize this is confusing and have now provided a better explanation of this data in both the Results and Methods sections of the manuscript (page 8, lines 168-170 and page 15, lines 310-314).

6. At the moment it is not clear if the loss of *Basp1* only results in de-repression of *Lef1* and *Ptch* expression or if there is an active shift towards activation of the genes. This good be test rather easily and provide much information on the mechanism behind this shift. Essafi et al showed that the difference in *Wt1* acting as suppressor or activator is mirrored in the binding of *Basp1* or CBP/p300 to the *Wt1*-bound enhancer element. Determining whether CBP/p300 binding increases upon *Basp1* loss would determine if there is an active shift towards an earlier stage or not.

We agree with the Reviewer that determining how chromatin remodeling factors such as CBP/p300 contribute to the *Lef1* and *Ptch1* expression changes will be of considerable interest. The work of ourselves and others (including Essafi et al.) in model cell lines has led to a working model in which BASP1 displaces CBP/p300 and recruits Histone deacetylase I, precipitating the alteration of histone marks at H3K9 and H3K27 (Essafi et al., 2011; Toska et al., 2012, 2014). A thorough analyses of the complex chromatin remodeling events stimulated by BASP1

is ongoing in the laboratory. We feel that the breadth of these effects forms a new study. However, we appreciate the interest in this point and have now mentioned it in the discussion section.

7. I assume Krt8 is not specific for only the taste buds. Could the authors comment on (potential) phenotypes in other organs / cell types?

Krt8 is expressed in epithelial cells at other sites in the adult mouse including the small intestine, bladder, liver, kidney and mammary glands (Zhang et al., 2012, Transgenic Res. 21, 1117-1123). Unfortunately, a detailed analysis of BASP1 expression is lacking and further work will be required to determine the extent of overlap of Krt8 and BASP1 expression. Our previous work has shown that BASP1 is expressed exclusively in the podocyte cells of the adult kidney. Krt8 is only expressed in the kidney tubule epithelia and not in podocyte cells of the adult.

8. Has the expression of Basp1 been studied in other tissues, and if so, is its expression always linked to Wt1 or could Basp1 have additional Wt1-independent roles?

BASP1 has been mostly studied in the brain which is not a site of Krt8 expression. Although BASP1 is expressed in many cell types along with WT1 (Carpenter et al., 2004), it is not exclusively linked with WT1. BASP1 can modulate the transcriptional repression function of ER α . Importantly, our previous work (Shandilya et al. Cell Death Dis. 2016 7:e2433) has demonstrated that ER α is not expressed in taste cells.

Reviewer 2 comments

WT1 appears to be expressed in nucleus in Fig. 1A, but not so obvious in Fig. 1B or 1D. I am wondering how specific the antibody against WT1 is. In the manuscript, the authors cited Toska et al for validation of antibody specificity. A bit more details should be provided to make this clear.

The Reviewer refers to Fig 2. We now provide control panels for the immunohistochemistry in the new Fig S1. We used the same WT1 antibodies as previously in our study of the WT1 knockout model (Gao et al., 2014) and the staining patterns of the CV are consistent with the current manuscript. In our previous and current study WT1 shows both nuclear and cytoplasmic localization and we now specifically mention this in the manuscript. As mentioned in the response to reviewer 1 we now include a diagram of the CV structure in Fig 2A to assist in explanation.

For Fig. 3D, is there a particular reason using Chi-square for statistical analysis? How many cells were sampled?

Chi-square was used for statistical analysis because we wanted to determine if the loss of BASP1 affects the number of taste cells that respond to a particular stimuli. That gives us a measure of the overall responsiveness of the taste cells. We also measured the peak amplitude of the remaining responses which identifies any changes in the evoked responses that occur. Taken together, these data demonstrate that the overall ability of the cells to respond to taste stimuli is significantly impaired by the loss of BASP1 and that the cells which are able to

respond are not responding as well as the control cells to the bitter and sweet stimuli we used. There was no significant difference in the size of the cellular response to 50mM KCl, even though significantly fewer cells responded in the BASP1-KO mouse. The number of cells sampled is now included in the figure legend.

It is unclear from the method section about how RNA was extracted from taste cells. More details will help readers to know what proportions of the preparation are from taste tissue and from the surrounding non-taste tissue. How did the authors control this for their experiments?

We have now included more information on the isolation of CV taste cells from the mice in the methods section of the manuscript. Specifically, taste cells are isolated using our standard taste cell isolation protocol which we routinely perform. We do not believe epithelial cell contamination is an issue as we target our cell collection to the taste cells that can be identified by their presence in the taste bud. If epithelial cells are present, it will be very low amounts.

The authors draw their conclusion in a large part by immunostaining. One caveat is whether the staining for the same marker was done side by side for sections from control and knockout mice. This potentially could influence the relative intensity of staining.

All comparisons between wild type and knockout tissues were performed side by side. This is now clearly stated in the methods section of the manuscript.

Reviewer 3 comments

Figure 1 From the images included in A and B it is not possible to identify double labeling. The authors do not indicate if these micrographs were obtained using confocal or light microscopy. If the latter, then double labeling structures simply cannot be reliably determined. If the former, the authors need to show single optical sections or indicate how many optical sections are compressed in each image. Further in the methods, the authors do not include criteria they used to classify double immunostaining. The quality of the microscopy in panel B is poor, especially the top 2 panels of the right column are not of sufficient resolution.

The images in Figure 1A and 1B were obtained using confocal microscopy and show compressions of 5 optical sections, as were all of the images in the manuscript. We regret our oversight in not indicating that previously. In response to requests by Reviewer 1 we have included additional timepoints and have also included new images of higher quality. We have removed the higher magnification images since specific localization of BASP1 within the taste bud is addressed in Figure 2. The main conclusion from Figure 1 is that BASP1 expression is confined to the gustatory nerve in the initial stages of development and localizes to cells within the taste buds during the end stages.

Figure 2. comments for Figure 1 apply, re: clear information of microscopy and criteria for double immunostained cells is lacking. The authors state for Fig 2A that: " it (WT1) shows a large degree of colocalization with BASP1." Please provide quantification to support this statement. Same critiques for panels B and C. Panel E. How is nuclear

localization of BASP1 determined? The authors need a nuclear counterstain to make this judgement. The authors state that 7/27 Shh+ cells had nuclear BASP1. How many taste buds from how many mice were used?

We have made the descriptions of the data in Figure 2 clearer. Both WT1 and BASP1 show nuclear and cytoplasmic staining as assessed by comparison to DAPI (new Fig 2C). We now state in the number of mice and taste buds that were analyzed in the text. Due to the dynamic nature of the taste cell renewal process in the taste buds, it is not always possible to detect Shh+ cells in a given taste bud. At most 1-2 positive cells (as shown in the figure) are detectable.

Figure 3. In the BASP1 conditional mice it appears that BASP1 expression is knocked down, not not knocked out, at both the protein and mRNA levels. The authors should use "knockdown" throughout. Please clarify if qPCR results reflect levels of BASP1 in the entire CVP epithelium or from plucked taste buds. Please indicate in the legend the number of mice and number of cells that contributed to the histograms in D and E.

This point was also raised by Reviewer 1 (comment 2). Since taste cells are renewed with an average half-life of 8-12 days there will be some replacement of BASP1 expressing cells in the 7 days following tamoxifen cessation before the tissue is isolated. We therefore appreciate the Reviewers point and describe BASP1 expression as "knocked down" throughout.

The qPCR was performed on RNA isolated from isolated taste buds. The isolation of taste cells is routinely performed in our laboratory and we are confident that the RNA is isolated from preparations of taste cells that contain a low level of contaminating epithelial cells.

Figs 4 and EV3. BASP1 knockdown in Krt8 taste cells causes reductions in some markers of type II and III cells but not others, and that their expression data correlate with reduced responses of isolated taste cells to sweet and bitter and KCl depolarization. However, gustducin is expressed by bitter detecting cells in the CVP but expression levels are not changed. The data don't support the general conclusion the authors make at the end of the first paragraph on page 8.

We showed the gustducin expression data to demonstrate that BASP1 knockdown does not cause a general deregulation of all taste cell specific proteins. PLC β 2 is required along with gustducin for bitter detection. Thus, the reduction in bitter detection by BASP1 knockdown cells can be explained by reduced PLC β 2 expression. We have now made this clear in the manuscript.

Methods: the authors state that "antibody specificity was verified as described in Toska et al., 2014". This paper uses human cell lines, not mice and therefore they need to include sections processed with no primary as negative controls for immunostaining in mice.

We have now included the negative controls suggested by the reviewer. We now also reference Gao et al (2014) for analysis of WT1 in mouse tissue and Green et al (2018) which examined a mouse cell line.

May 17, 2019

RE: Life Science Alliance Manuscript #LSA-2018-00287-TR

Dr. Kathryn F Medler
University at Buffalo
Biological Sciences
University at Buffalo
Cooke Hall
Buffalo, NY 14260

Dear Dr. Medler,

Thank you for submitting your revised manuscript entitled "The WT1-BASP1 complex is required to maintain the differentiated state of taste receptor cells". Two of the original reviewers have re-evaluated your manuscript again. As you will see, the reviewers appreciate the introduced changes and only point to further text changes needed. We would thus be happy to publish your paper in Life Science Alliance pending final revisions:

- Please address reviewer #2's remaining comment
- Please provide the manuscript text as a word docx file
- Please indicate which statistical test has been used throughout all figures (currently already done once and in methods section)
- Please note that Fig 2G is misspelled as panel E in the figure legend
- Figure 1 spreads over two pages, please provide this figure on a single page
- Please note that we display supplementary figures in-line in the HTML version of the paper - please upload all supplementary figures as individual files and without legends, the legends can go into the main manuscript file
- Please link your profile in our submission system to your ORCID iD, you should have received an email with instructions on how to do so

A. FINAL FILES:

B. MANUSCRIPT ORGANIZATION AND FORMATTING:

Sincerely,

Andrea Leibfried, PhD
Executive Editor
Life Science Alliance
Meyerhofstr. 1

69117 Heidelberg, Germany
t +49 6221 8891 502
e a.leibfried@life-science-alliance.org
www.life-science-alliance.org

Reviewer #1 (Comments to the Authors (Required)):

The authors have addressed my comments appropriately. Although the primers indicated in fig S2 are not arm spanning (they span the deletion, but do not test for correct targeting), with the better antibody controls and good explanation for the remaining signal I'm satisfied with the overall description of the model. The schematic in Fig 2A is certainly appreciated by this reviewer and makes things a lot clearer.

In all I congratulate the authors with a nice piece of work.

Reviewer #2 (Comments to the Authors (Required)):

The authors have largely addressed my comments.

However, I would think the use of "knock down" in the revised manuscript is misleading because this is a conditional knockout approach. The reduced immunostaining signal could be due to either non-specific staining of the antibody use (secondary alone cannot address this), or residual protein after the gene being ablated in Krt8+ cells. Also, Krt8 is not expressed in every mature taste cells. Some of cells showing a bit strong signal could be due to this reason.

Therefore, I would recommend the authors to reword the sentences where "knock down" is used because it is misleading.

May 27, 2019

RE: Life Science Alliance Manuscript #LSA-2018-00287-TRR

Dr. Kathryn F Medler
University at Buffalo
Biological Sciences
University at Buffalo
Cooke Hall
Buffalo, NY 14260

Dear Dr. Medler,

Thank you for submitting your Research Article entitled "The WT1-BASP1 complex is required to maintain the differentiated state of taste receptor cells". It is a pleasure to let you know that your manuscript is now accepted for publication in Life Science Alliance. Congratulations on this interesting work.

DISTRIBUTION OF MATERIALS:

Again, congratulations on a very nice paper. I hope you found the review process to be constructive and are pleased with how the manuscript was handled editorially. We look forward to future exciting submissions from your lab.

Sincerely,

Andrea Leibfried, PhD
Executive Editor
Life Science Alliance
Meyershofstr. 1
69117 Heidelberg, Germany
t +49 6221 8891 502
e a.leibfried@life-science-alliance.org
www.life-science-alliance.org